# Intraspecific variability in plant and soil chemical properties in a common garden plantation of the energy crop *Populus*

**Matthew E. Craig[1], Anne E. Harman-Ware[2], Kevin R. Cope[3], Udaya C. Kalluri[3]***

**1** Environmental Sciences Division, Oak Ridge National Laboratory, Oak Ridge, TN, United States of America, **2** Renewable Resources and Enabling Sciences Center, National Renewable Energy Laboratory, Golden, CO, United States of America, **3** Biosciences Division, Oak Ridge National Laboratory, Oak Ridge, TN, United States of America

* kalluriudayc@ornl.gov

**Data Availability Statement:** All raw data associated with the study has been provided in a supplemental file [S2 Table_and raw data].

## Abstract

Optimizing crops for synergistic soil carbon (C) sequestration can enhance $CO_2$ removal in food and bioenergy production systems. Yet, in bioenergy systems, we lack an understanding of how intraspecies variation in plant traits correlates with variation in soil biogeochemistry. This knowledge gap is exacerbated by both the heterogeneity and difficulty of measuring belowground traits. Here, we provide initial observations of C and nutrients in soil and root and stem tissues from a common garden field site of diverse, natural variant, *Populus trichocarpa* genotypes—established for aboveground biomass-to-biofuels research. Our goal was to explore the value of such field sites for evaluating genotype-specific effects on soil C, which ultimately informs the potential for optimizing bioenergy systems for both aboveground productivity and belowground C storage. To do this, we investigated variation in chemical traits at the scale of individual trees and genotypes and we explored correlations among stem, root, and soil samples. We observed substantial variation in soil chemical properties at the scale of individual trees and specific genotypes. While correlations among elements were observed both within and among sample types (soil, stem, root), above-belowground correlations were generally poor. We did not observe genotype-specific patterns in soil C in the top 10 cm, but we did observe genotype associations with soil acid-base chemistry (soil pH and base cations) and bulk density. Finally, a specific phenotype of interest (high vs low lignin) was unrelated to soil biogeochemistry. Our pilot study supports the usefulness of decade-old, genetically-variable, *Populus* bioenergy field test plots for understanding plant genotype effects on soil properties. Finally, this study contributes to the advancement of sampling methods and baseline data for *Populus* systems in the Pacific Northwest, USA. Further species- and region-specific efforts will enhance C predictability across scales in bioenergy systems and, ultimately, accelerate the identification of genotypes that optimize yield and carbon storage.

**Funding:** This work was supported by funding from the Center for Bioenergy Innovation (CBI) project to UCK. CBI is a U.S. Department of Energy Bioenergy Research Center supported by the Office of Biological and Environmental Research in the DOE Office of Science. The funding agency played no role in the study design, data collection and analysis, decision to publish, or preparation of the manuscript. Oak Ridge National Laboratory is managed by UT-Battelle, LLC for the US DOE under Contract Number DE-AC05-00OR22725.

**Competing interests:** The authors have declared that no competing interests exist.

## Introduction

Enhanced soil carbon (C) storage—i.e. removing $CO_2$ from the atmosphere in favor of a longer-lived terrestrial pool—represents one possible nature-based climate solution [1, 2]. Agricultural systems—including bioenergy systems—have the greatest potential for enhanced soil C because these systems have been the most depleted due to human land use [3, 4]. Frontier technologies offer the potential to enhance soil C sequestration beyond what is currently achievable with best management practices [5], especially if combined with other strategies like BECCS (bioenergy with carbon capture and storage) [6]. One promising frontier approach is to optimize crops for synergistic soil C sequestration [7]. This involves identifying candidate genotypes that associate with soil C promoting phenotypes [8] or, better, genotypes that associate with positive soil C outcomes (e.g. net accumulation). Detecting genotype-soil C associations is challenging because there are few resources suitable to answer this question, and because soil biogeochemistry changes slowly and via many mechanisms in response to plant trait variation.

Long-term (e.g. >10 years old) plant genetic diversity panels, planted in common gardens for genome wide association studies [9], offer an opportunity for unprecedented studies of associations between plant genotypes and soil C dynamics. Several such plantations exist for the bioenergy candidate, *Populus trichocarpa* [10]. However, in these systems, genotypes are intermixed and individual trees are planted in 1–2 m apart [11]. Studying effects of individual genotypes on soil properties, therefore, requires a focal tree approach, instead of the plot-level measurements that are typical in ecosystem-level soil biogeochemistry studies. This presents a potential methodological barrier because soil C and other biogeochemical properties are spatially heterogeneous at fine spatial scales [12], which can also drive fine-scale variability in plant traits [13]. Thus, prior to any broad-scale sampling efforts to leverage and inform plant-soil interactions in large genetic diversity plantation systems, there is a need for empirical evaluation of sampling strategies and to address whether plant-soil couplings are detectable, variable, and replicable at the scale of individual plants or genotypes.

The difficulty of measuring belowground plant traits presents an additional challenge for studying genotype associations with soil C and other properties. Belowground plant inputs, including root tissues and exudates, are now recognized as critical drivers of soil C formation [14–17], and soil microbial traits mediate both soil C formation and loss [18]. Thus, assessing belowground plant and microbial traits is critical to understanding soil C dynamics, especially in agricultural and managed forest systems where aboveground biomass is harvested. Focal tree root sampling is time consuming in these plantation systems because it requires tracing excavated roots to the genotype of interest. The challenging nature of sampling and measurement has therefore resulted in asymmetric knowledge gaps in belowground dynamics relative to heavily studied above-ground dynamics in bioenergy crop plantations. Inferring belowground traits from aboveground traits is one potential way around this issue. However, aboveground-belowground trait associations have mixed support among species [19], and not much is known about these associations among different field-cultivated genotypes for *P. trichocarpa*. In summary, *Populus* Genome-wide association study (GWAS) plantations represent a promising system for identifying important genotype-specific associations with belowground properties but given the challenges of working in such a system, an initial evaluation is critical to design informed sampling campaigns at scale.

In the present study, we leveraged a genetically diverse, long-term *P. trichocarpa* population —with a wide range of phenotypic variation reported in aboveground traits—growing in a common garden field setting in the Pacific Northwest, USA [20]. Replication of genotypes in blocks provides the opportunity for capturing genotypic effects, but belowground soil and

plant traits have not yet been studied in this population. We hypothesized that distinct *P. trichocarpa* genotypes determine variation in soil properties, and that this variation is observable at the scale of individual trees. We also hypothesized that plant lignin phenotype influences amount of C in soil and that belowground traits correlate positively with aboveground traits across genotypes. We evaluated these hypotheses using a woody perennial bioenergy plant in a ten-year old intraspecific GWAS site. Our objectives were to evaluate 1) the potential of focal tree soil sampling for detecting associations of poplar genotypes with soil biogeochemistry (e.g. C, nitrogen (N), phosphorus (P), and a range of soil micro- and macronutrient elements) and soil bulk density, and 2) the strength of correlations between above- and belowground traits (stem vs root and plant vs soil). To do this, we measured soil, root, and stem chemical traits for several focal trees spanning four different *P. trichocarpa* genotypes representative of population extremes in cell wall chemistry (Fig 1A)—i.e. phenotypes with low (BESC 24) or high (BESC 375, BESC 371, BESC 394) lignin content based on population-wide cell wall chemistry data previously reported for this population [13, 20–23]. We also explored the influence of sampling location on measured soil chemical traits, potential effects of plant tissue chemistry on soil C, relationships among nutrient concentrations among plant and soil pools.

## Materials and methods

### Field site description

The *Populus trichocarpa* GWAS Common Garden site field site is located in Clastkanie, Oregon, USA (coordinates: -123.26, 46.12). Sampling was conducted on this private land and research activities were permitted via Poplar Innovations Inc. through an ongoing access agreement with the owner. The site is flat with a relatively uniform soil type; Entisols [24] formed by floodplain activity adjacent to the Columbia River. Continuous land use in past decades has been for *Populus* plantations. For this study, we collected surface soil (top 10 cm) from planted areas and from areas that do not contain *Populus* individuals in the current plantation. Analysis of basic soil characteristics at the time of sampling revealed an average pH of 5.33, gravimetric soil moisture content of 31.3%, C:N ratio of 12.24, and soil texture as Silty Clay (10.0% sand, 47.5% silt and 42.5% clay). The common garden site has ~1000 genotypes of *P. trichocarpa*, densely planted (<2 m apart) and replicated in three blocks with randomization and surrounded by two buffer rows on all four edges to prevent edge effects. These trees had been growing for more than 11 years at the time of sampling.

### Field sample collection

Bulk soil samples (0–5 cm [top] and 5–10 cm [bottom] depth) were collected 30 cm from the focal tree trunk in three randomly chosen cardinal directions, digging using graduated trowels to reference depth and collecting and storing in labeled Ziploc bags. Two replicate trees were sampled for each of the four genotypes. For bulk density analysis, soil samples were collected using AMS Soil Core Sampler Kit with Hammer Attachment (part # 77455), 2 in × 2 in Soil Core Kit and stored in 2 in × 2 in core liners Precut (plastic). Samples were shipped on ice and stored at 4˚C until analysis. Prior to analyses, soils were processed using forced air drying for ~48 hours at 43˚C. Stem cores (1 cm in diameter) from tree breast height (~1.3 m) were collected using increment borer and stored and shipped in zip-lock bags at −20˚C. Fine roots (<2 mm diameter) were accessed by shallow digging, tracing of roots to tree stump and cut using ethanol wiped pruners and stored and shipped in ziplock bags on dry ice. We targeted the surface-most roots which usually corresponded to a depth of less than 10 cm from the surface. For elemental analysis, subsamples of roots and stems were separately aliquoted into labeled

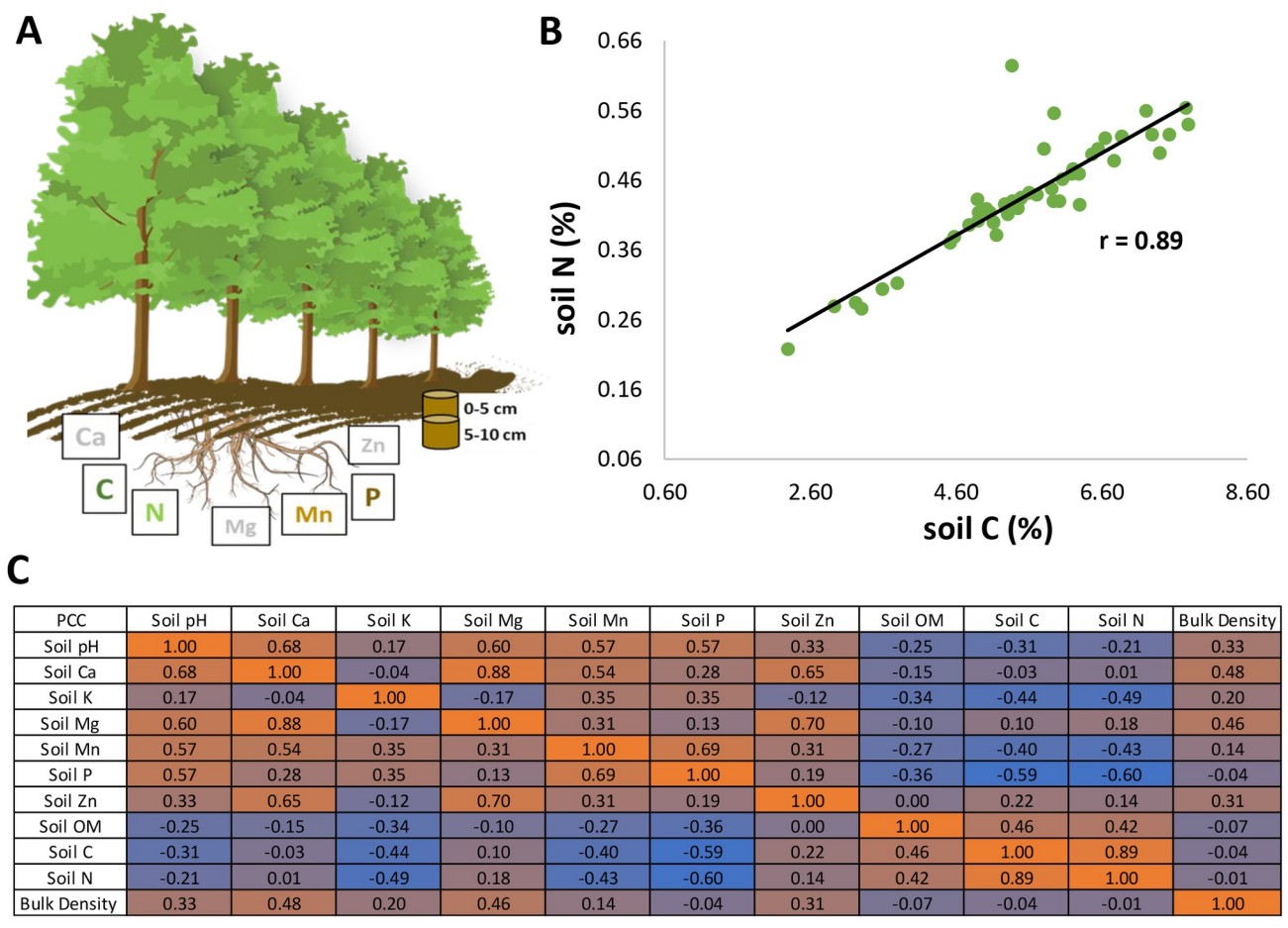

**Fig 1. Correlation analysis undertaken at the *Populus* common garden site at Clatskanie, OR.** (A) Graphical representation of plant-soil elemental distribution analysis undertaken in top 10 cm surface soil profile (bulk soil collected from 0–5 cm and 5–10 cm depth pools). (B) Bivariate relationship between C and N using all soil samples (r = 0.89). (C) Correlation matrix for all soil properties measured. Twelve observations for each genotype.

brown paper bags and dried in hot air oven 70°C for three days and shipped to soil service lab in University of Georgia.

## Laboratory analyses

Basic soil analysis and elemental analysis of plant and soil samples was conducted at the soil testing service of UGA Extension, Agricultural and Environmental Services Laboratories using standard methods. Soil pH (1:1$_{w:v}$ soil:0.01 M CaCl$_2$) and carbon and nutrient concentrations were determined using published standard methods [25, 26]. Briefly, soils were dried, ground, and analyzed for various macro and micronutrient elements; carbon and nitrogen (N) on an elemental analyzer [27, 28]; Phosphorus (P), Potassium (K), Calcium (Ca), Magnesium (Mg), Zinc (Zn), and Manganese (Mn) were extracted in a Mehlich-1 solution and analyzed via Inductively Coupled Plasma-Optical Emission Spectrometry (ICP-OES). Soil texture analysis was performed using the hydrometer method [29]. Stem and root C and N were analyzed in the same manner as the soils. Additionally, concentrations of macro and micronutrients were determined via ICP-OES after microwave digestion in nitric acid.

## Data analysis

To examine the extent to which soil chemical traits vary at the scale of individual trees, we fit linear models with focal tree and sample depth as predictors (R 3.6.1). Assumptions of normality and homogeneity of variance were checked by inspecting residual plots and with a Shapiro-Wilk test. Response variables were natural log-transformed where necessary to improve adherence to assumptions (specifically for soil C and N content). To determine the proportion of variance explained at the level of individual trees, partial $R^2$ were determined using the 'asbio' package [30]. To provide an initial look at whether soil variables are clustered with respect to genotype, we additionally calculated partial $R^2$ for models fit with genotype and sample depth as predictors. However, these analyses are considered exploratory due to inadequate replication.

Relationships, principal components and variance amongst stem, root and soil traits (elemental, chemical and physical properties) were measured by determination of Pearson Correlation Coefficients (PCC) and Principal Component Analysis (PCA). PCC values were calculated on raw data values and PCA (100 iterations of NIPALS algorithm, 20 random cross validation) was performed on mean-centered compositional data using the Unscrambler X V.10.5 (Camo Analytics, AspenTech).

## Results

Analyses of field-collected plant and soil samples undertaken to assess whether genetically diverse, long-term (ten-year) stands of a natural *Populus trichocarpa* collection which are proven resources for aboveground studies, present a practical resource for quantifying genotype-specific plant effects on soil chemical properties, including soil C (Fig 1A). If so, studying such systems could facilitate the deeper probing and optimization of poplar plantation systems for enhancing C storage in soils. Our study system is well suited to address this question because soil biogeochemical properties change slowly, and different genotypes are intermixed within replicated blocks on the field site allowing insights into feasibility and extent thereof of studying aboveground-belowground correlations at plant, genotype, and population levels in this system.

### Variability in soil characteristics at the Clatskanie field site

Across the entire field site soil, the correlation between soil C and soil N was strong (r = 0.89; Fig 1B), demonstrating a consistent soil C:N ratio. Correlation analysis of all measured soil parameters (pH, bulk density, and elements) showed additional strongly correlated soil properties (Fig 1C). PCA analyses conducted to explore whether there is a dependence of overall soil chemistry on either the radial direction (East, North and West) or depth (top [0–5 cm] vs bottom [5–10 cm]) of sampling showed no significant effect overall (S1 Fig). However, linear models did show an association with sample depth ($p < 0.05$) for specific soil variables—pH, Ca, K, Mg, and P (S1 Table)—with nutrient concentrations and pH tending to be higher in the top surface layer (e.g. Ca in Fig 2A).

### Focal tree and genotype-specific plant-soil relationships

Data from the pilot study support that soil properties can cluster at the genotype or individual tree level (S2 Fig). Focal tree was a significant predictor of all analyzed soil properties ($p < 0.001$), and explained more than half of the variation ($R^2_{partial} > 0.50$) in all properties except for soil C:N ($R^2_{partial} = 0.50$; S1 Table). For models using data aggregated at the scale of genotype, variables related to soil acid-base chemistry clustered together for different

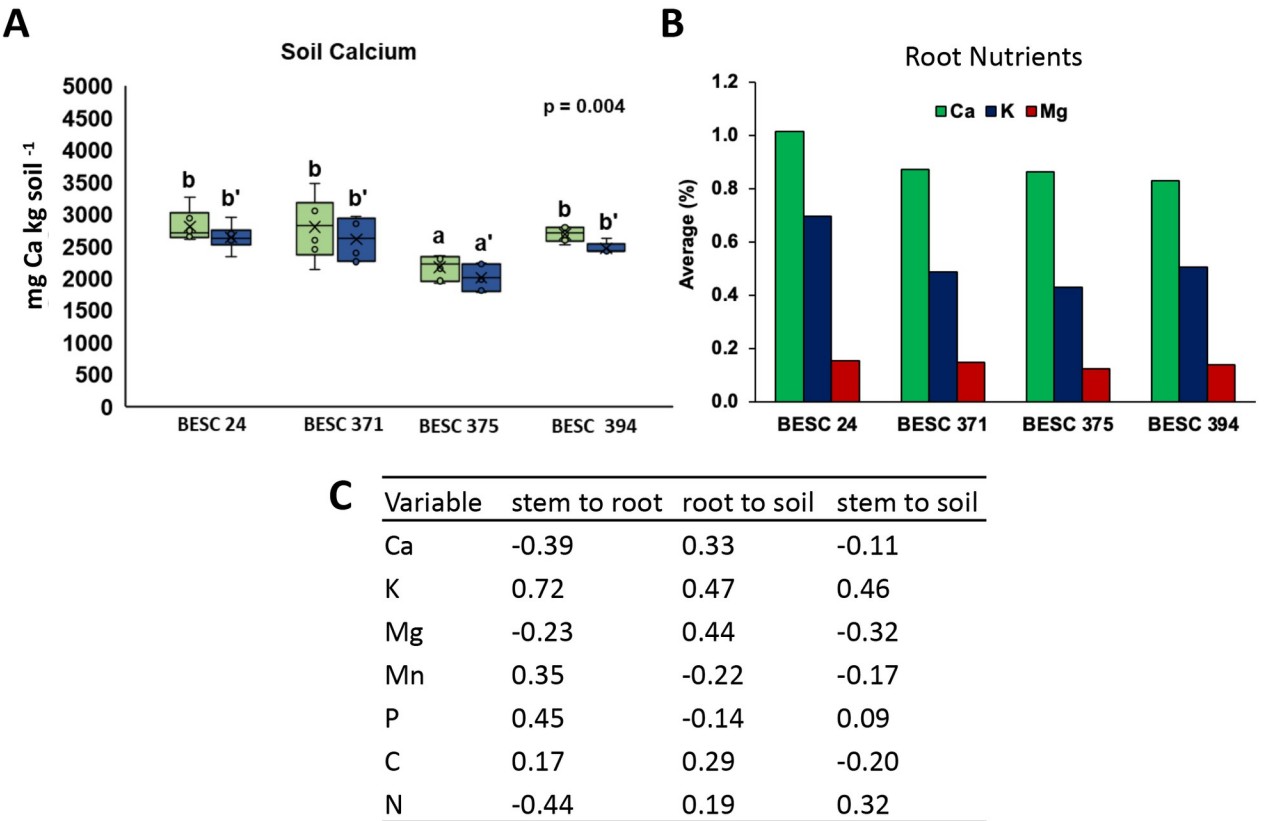

**Fig 2. Representative observations of elemental distribution with genotype, within tissue type and across above and belowground samples.** (A) Ca concentration in soil collected from BESC 375 from across the replicate blocks was observed to be significantly different relative to other genotypes based on ANOVA ($p < 0.05$). Horizontal bars indicate significant differences between 0–5 cm (green) and 5–10 cm (blue) soil samples. (B) Within the same plant tissue (root), elemental distribution (shown here Ca, K, and Mg) varied across genotypes. (C) Pearson's correlation coefficients among stem, root, and soil concentrations for carbon and key nutrients. Bars and boxes in panels A and B represent three observations from each of two individuals. Coefficients in (C) represent correlations among average values for six individuals. A comprehensive correlation matrix generated using all parameters measured for stem, root, and soil samples obtained from the four genotypes is provided in S2 Table.

genotypes. Specifically, genotype predicted more than half of variation in pH and Ca and predicted more than a quarter of variation in Mg and Mn (S1 Table). Visual inspection of plots reveals that one genotype, BESC 375, is a strong driver of these patterns (S2 Fig) and was associated with more acidic (low pH), calcium-depleted soil (Fig 2A).

PCA analysis comparing the genotype representing the low end of lignin distribution in the population (BESC 24) with a high lignin genotype (BESC 375) showed genotype-specific clustering of soil properties (S3 Fig), but this lignin phenotype based differential clustering was not observed when all four genotypes were included in the analysis. BESC 24 did stand out when comparing root elemental composition across genotypes (Fig 2B), having the greatest concentration of Ca, K, and Mg (important soil base cations), but also the highest root C:N ratio (62.9 versus 49.9–59.2 for the other genotypes). Lastly, we observed associations between plants and soil physical properties. BESC 24 tended to associate with higher bulk densities in the deeper measured soil layer. Moreover, *Populus* presence, regardless of individual, genotype, or sample

depth, was associated with significantly lower soil bulk density compared with a control micro-site i.e., a location containing no *Populus* individual within a 5 m radius and with no plant roots evident in the soil (S4 Fig).

### Extent of correlation among aboveground-belowground chemical traits

We found limited evidence to suggest coordination of stem and root traits among plant geno-types (S2 Table). In general, nutrient concentrations were not correlated across tissues (stem-root) or between tissues and soil (stem-soil, root-soil; Fig 2C). One possible exception to this is stem and root K concentrations, which were found to be positively correlated (PCC = 0.72). A comprehensive correlation matrix generated using all parameters measured for stem, root and soil samples obtained from the four genotypes is provided as S2 Table. Some correlations among different elements within a pool were detected. For example, soil bivalent cations ($Ca^{2+}$ and $Mg^{2+}$) were positively correlated and were positively correlated with soil pH (S2 Table). Correlations were also observed across pools and elements. For example, soil N was negatively related to root K and stem K (S2 Table).

### Discussion

Insights from our pilot study represent a first report on aboveground-belowground chemical trait correlations for different *P. trichocarpa* genotypes grown in a genetically diverse planta-tion. In general, our results support our first hypothesis that *P. trichocarpa* genotypes differen-tially associate with soil biogeochemical properties, and that this is observable at the scale of individual trees (Fig 2 and S2 Fig). The pronounced variability in soil biogeochemical proper-ties across our common garden indicates that planted trees are likely driving differences in soil C and nutrient cycling, though we note that there could be other sources of soil heterogeneity (e.g. variability in microbial community or soil physical properties), even in a common garden. We found that most of the variation in soil biogeochemical properties in this system occurred among individual focal trees, exceeding micro-scale variability within each focal tree location (S1 Table). Typical common garden studies of tree effects on soil biogeochemistry employ a plot-based approach and are designed to detect changes over a larger area seeded with a com-mon species [31]. Fine-scale heterogeneity could make it difficult to detect species effects at a smaller spatial grain. For example, working in natural forests, Fraterrigo et al. [32] measured high variability in soil chemical properties within 20 × 20 m plots, but found that previous land use generally had a homogenizing effect at this scale. It is likely that previous land use has also led to reduced fine-scale soil heterogeneity in our study site. Regardless, our finding that soil biogeochemical properties clustered at the scale of our focal tree sampling indicates that our site, and similar plantations, could be used to study genotype-specific effects on soil biogeochemistry.

Our analysis showed that *Populus* genotypes could differentially affect soil biogeochemical properties, specifically, soil pH and soil base cations (except for potassium) were consistently lower for one genotype, BESC 375 (Fig 2 and S2 Fig). A number of traits could lead to differ-ences in acid-base chemistry under different strains of poplar including association with ecto-versus arbuscular mycorrhizal fungi, N or Ca uptake mechanisms, or organic acid production [33–35]. Though we did not find evidence for genotype effects on soil C in this limited sam-pling effort, we note that these effects on soil acid-base chemistry could indicate or lead to effects on soil C cycling. For example, soil Ca availability could enhance mineral-associated C formation via cation bridging mechanisms [36], an effect which likely depends on soil pH [37]. Changes in soil pH have also been shown to correlate with decomposition rates [38], soil microbial community composition [39], and mineral-organic reactivity [40], among other

factors. Thus, soil pH alterations may be a bellwether for soil C effects. Building upon the merit of studying impacts of intraspecies variability on associated soil properties, in the future, a more comprehensive probing that includes C forms (i.e., mineral associated or particulate organic C) and microbiome analyses across a depth and temporal profile will be useful in developing prediction frameworks. Most importantly though, our findings of differences among focal trees and trends toward genotype differences pave the way for a more comprehensive and well-replicated comparison of genotype-specific effects in this system.

Contrary to our second hypothesis, we did not find evidence of consistent aboveground-belowground correlations for tissue chemistry traits among our study individuals (Fig 2; S2 Table). Plant economic spectrum theory predicts that leaf, stem, and root traits are correlated [41], a prediction that has received support across a range of ecosystems [42–44]. However, evidence to the contrary suggests that the root economic spectrum may be multidimensional, potentially leading to a breakdown of aboveground-belowground trait correlations [45, 46]. Thus, the lack of significant couplings between root and stem traits observed under field settings—which is also supported by results obtained from independent greenhouse observations of the natural variant population (Kalluri et al. unpublished data)—indicates a need to collect both aboveground and belowground phenotypic data in future sampling efforts in this system to further evaluate this hypothesis.

## Conclusions

Overall, our assessment supports the utility of long-term, genetically diverse bioenergy crop plantations to study plant-soil relationships at the scale of individual trees and genotypes. Our results show substantial variation in soil biogeochemical properties at the scale of individual trees and provide evidence of genotype-specific effects on soil acid-base chemistry. Though select correlations between plant and soil properties were detected, we generally observed poor correlations among above- and belowground chemical traits. Additionally, variability in poplar genotype showed a stronger association with soil biogeochemistry than the phenotype on which our sampling was based (i.e., high vs low lignin). Findings from the present proof-of-concept pilot study show that evenly spaced, genetic diversity plantations on relatively uniform field topology and basic soil properties can be useful for probing and identifying candidate genotypes and genes based on their effects on key soil biogeochemical properties. Such systems can contain hundreds of distinct genotypes enabling GWAS-scale investigations of soil biogeochemical properties. We conclude that these reported plant-soil relationships should be studied more comprehensively considering deeper soil profiles, more genotypes, mineral and particulate associated soil organic C fractions, and microbiome diversity, to facilitate comprehensive measurement and modeling of both the productivity and uniformity of aboveground biomass in bioenergy systems and the effects of bioenergy crop production on soil health and carbon sequestration.

## Supporting information

**S1 Fig. PCA analysis.** PCA analysis of all soil samples (n = 48) collected from variable depths and radial direction of sampling using compositional analysis traits. PC-1 explains 95% of variance is driven primarily by Ca content while PC-2 corresponds to variation in K content. (DOCX)

**S2 Fig. Soil chemical property variation across individuals and genotypes.** Boxplots showing pH, nutrient concentrations, and C:N in soils sampled at each of eight focal trees belonging to four different *Populus trichocarpa* genotypes (delineated with shading). The two replicates

of each genotype (BESC 24, BESC 371, BESC 375, BESC 394) are indicated with the suffix A and B along the x-axis. Boxes represent three observations per individual. Units for carbon (C) and nitrogen (N) are percent by mass. Soil pH (pH) and soil C-to-N ratio (CN) are unitless and the remaining nutrients are expressed in mg kg$^{-1}$.
(DOCX)

**S3 Fig. Targeted PCA analysis.** PCA analysis of chemical properties of all soil samples collected from 0–10 cm depth for BESC 24 (low lignin) and BESC 375 (high lignin). Twelve observations for each genotype.
(DOCX)

**S4 Fig. Bulk density data.** Bulk density data corresponding to soil cores obtained from (Blue) "Top" soil cores [soil cored at the 0"-2" depth from surface of the soil horizon] and (Green) "Bottom" soil cores [soil cored at the 2"-4" depth from surface of the soil horizon]. Four observations per genotype included two observations each for "top" or "bottom" cores.
(DOCX)

**S1 Table. Variance in soil biogeochemical properties.** Variance in soil biogeochemical properties explained (Partial $R^2$) by linear models containing either individual trees (n = 3 soil samples per tree) or genotypes (2 trees per genotype, n = 6). Individual trees were significantly related to soil properties ($p < 0.001$) for all measured variables. No statement about statistical significance is made about genotype effects due to insufficient replication at this level of analysis. A significant effect of sample depth is denoted with an asterisk (*) for response variable.
(DOCX)

**S2 Table. Comprehensive correlation matrix.** Comprehensive correlation matrix presented as a heatmap generated using PCC of all parameters measured for stem (two observations per genotype), root (two observations per genotype) and soil (12 observations per genotype) samples obtained from the four genotypes.
(XLSX)

## Acknowledgments

We thank Drs. Eric Pierce and Gerald Tuskan for early consultations on the field sampling campaign, Sara Jawdy at Oak Ridge National Laboratory for sample aliquoting and shipment to soil service center, Kat Haiby, Brian Stanton and Rick Stonex at Poplar Innovations Inc. for facilitating field site access, and UGA soil service center for analysis of the study samples.

## Author Contributions

**Conceptualization:** Anne E. Harman-Ware, Udaya C. Kalluri.

**Data curation:** Matthew E. Craig, Kevin R. Cope.

**Formal analysis:** Matthew E. Craig, Anne E. Harman-Ware, Kevin R. Cope.

**Funding acquisition:** Udaya C. Kalluri.

**Investigation:** Anne E. Harman-Ware, Udaya C. Kalluri.

**Methodology:** Anne E. Harman-Ware.

**Project administration:** Udaya C. Kalluri.

**Resources:** Udaya C. Kalluri.

**Supervision:** Udaya C. Kalluri.

**Visualization:** Matthew E. Craig, Anne E. Harman-Ware, Kevin R. Cope.

**Writing – original draft:** Matthew E. Craig, Anne E. Harman-Ware, Udaya C. Kalluri.

**Writing – review & editing:** Matthew E. Craig, Anne E. Harman-Ware, Kevin R. Cope, Udaya C. Kalluri.

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
