## [Decision Letter · Decision Letter 0]

28 May 2024

PONE-D-24-05944Intraspecific variability in plant and soil chemical properties in a common garden plantation of the energy crop PopulusPLOS ONE

Dear Dr. Kalluri,

Thank you for submitting your manuscript to PLOS ONE. After careful consideration, we feel that it meets PLOS ONE’s publication criteria after miner revisions. Therefore, we invite you to submit a revised version of the manuscript that addresses the points raised during the review process.please go through figures and make sure the the colors can be distingushed. The choice of dark purple is difficult to tell from the dark blue in the second figure. I also suggest that sample sizes are indicated in the legends throughout

We look forward to receiving your revised manuscript.

Kind regards,

Benedicte Riber Albrectsen

Academic Editor

PLOS ONE

Journal Requirements:

Reviewers' comments:

Reviewer's Responses to Questions

**Comments to the Author**

1. Is the manuscript technically sound, and do the data support the conclusions?

Reviewer #1: Yes

Reviewer #2: Yes

2. Has the statistical analysis been performed appropriately and rigorously? 

Reviewer #1: Yes

Reviewer #2: Yes

3. Have the authors made all data underlying the findings in their manuscript fully available?

Reviewer #1: Yes

Reviewer #2: Yes

4. Is the manuscript presented in an intelligible fashion and written in standard English?

Reviewer #1: Yes

Reviewer #2: Yes

5. Review Comments to the Author

Reviewer #1: Very interesting research in dependence (connection) and mutual influence between soil and plant through quantitative analysis of micro and macro elements. The authors set up the experiments well, emphasizing individual and genotypic differences in one species of poplar (Populus trichocarpa). Numerous collected results mainly prove individual variability (therefore also genotypic) significant dependence for most of the observed properties between soil and plant, but also a weak correlation for properties of the above-ground and underground parts of the tree.

I recommend that the paper be published with a few questions for the authors to think about or possibly supplement the text in the paper:

1) Is the observed interdependence of properties a result of the plant's influence on the soil or vice versa?

2) Are there any available data regarding soil properties before the planting of these genotypes, or were they exclusively taken from the control plot?

3) Should all of this be verified through controlled (closed) experiments, such as in a greenhouse using potted substrate?

Reviewer #2: Intraspecific variability in plant and soil chemical properties in a common garden plantation of the energy crop Populus

The study on the effects of Populus genotypes on physiochemical properties of soil, C accumulation and elemental content in shoots and roots is interesting. The concept of the study to reduce atmospheric carbon dioxide content through candidate plant genotype that promotes soil C accretion and plant phenotype is important in climate change context. The manuscript is well drafted, results are well explained. However, there are some corrections that the authors have to go through it.

1. Line 38 Specify plant tissues as roots and stem.

2. Line 41-42 Modify the line “the potential for co-optimizing above-belowground plant systems”.

3. Line 46 The above and below ground properties have not measured in the study. The authors have only measured nutrient composition so instead of “property” the nutrients/chemical components/elements” words are more appropriate.

4. Line 107 The study is not focused on the microbial traits/diversity, so I think donot mention it here because the paragraph is mainly directing the hypothesis and objectives of the study.

5. Line 117 -122 The hypothesis of the study should be mentioned first followed by the objectives. Moreover, the hypothesis should be in sentence form rather than in points.

6. Line 120-122 is the part of objective not hypothesis so rearrange it.

7. Line 107-122 Needs to be rewritten based on the above comments.

8. Line 128-132 Did the authors collect the soil samples from the plain land which was earlier used for populus plantation. If this is the case the land surface should be uniform throughout then what the authors are meant by under the planted and non-planted areas. And if the two areas differ in terms of plantation, then I am wondering how the soil properties should be same.

9. With the under-plant samples did the author means rhizosphere samples.

10. Line 138 How these soil samples differ from the samples mentioned in the above paragraph (line 128-132)

11. Line 140 Two replicates are very few, atleast five replicates should be taken for the good results.

12. Line 145 For tree breast height. Mention the exact height from the surface also.

13. Line 146. Mention the depth from the surface at which the root samples were collected.

14. Line 203 The text did not match Fig. 2A. According to the figure the Ca content is higher at 5-10 cm in all the genotypes not on the soil surface. Check it.

15. Line 220 -221 It is better if the increasing results of Ca, Mg and K in BESC 24 is shown in the % increase compared to other genotypes.

16. Line 229 Replace properties with chemical traits see the above comment 3.

17. Above-ground not only includes stem but leaves too then why the authors have not measured the chemical constituents in leaves.

18. Line 242 see the above comment on property.

19. Insert figure/table number in the discussion section while discussing your results so that it is easy for the readers to understand.

20. Line 295 see the above comment on property

6. PLOS authors have the option to publish the peer review history of their article (what does this mean?). If published, this will include your full peer review and any attached files.

Reviewer #1: **Yes: **Milan Mataruga

Reviewer #2: **Yes: **Lovely Mahawar

---

## [Author Response · Author response to Decision Letter 0]

6 Aug 2024

Dear Dr. Albrectsen,

Thank you for your letter of 28 May 2024 regarding our manuscript “Intraspecific variability in plant and soil chemical properties in a common garden plantation of the energy crop Populus”. We appreciate the positive response to our paper and the constructive and thoughtful comments from you and from the peer reviewers. We have carefully addressed each comment and now present a revised manuscript. 

Attached please find a word file named "Response to Reviewers" wherein review comments in listed bold, followed by our point-by-point response to each reviewer comment. Revisions made to the main document are in red font, and line numbers in our responses refer to those in our revised manuscript. We have additionally made changes to format the manuscript in accordance with PLOS ONE guidelines. Please do not hesitate to contact me if you have any questions. 

Sincerely, 

Udaya Kalluri (on behalf of all co-authors)

---

## [Editor Report · Decision Letter 1]

9 Aug 2024

Intraspecific variability in plant and soil chemical properties in a common garden plantation of the energy crop Populus

PONE-D-24-05944R1

Dear Dr. Kalluri,

We’re pleased to inform you that your manuscript has been judged scientifically suitable for publication and will be formally accepted for publication once it meets all outstanding technical requirements.

Kind regards,

Benedicte Riber Albrectsen

Academic Editor

PLOS ONE
---

## [Editor Report · Acceptance letter]

20 Aug 2024

PONE-D-24-05944R1 

PLOS ONE

Dear Dr. Kalluri, 

I'm pleased to inform you that your manuscript has been deemed suitable for publication in PLOS ONE. Congratulations! Your manuscript is now being handed over to our production team.

Kind regards, 

on behalf of

Dr. Benedicte Riber Albrectsen 

Academic Editor

PLOS ONE